# The Genomic Landscape of Colorectal Cancer in the Saudi Arabian Population Using a Comprehensive Genomic Panel

**DOI:** 10.3390/diagnostics13182993

**Published:** 2023-09-19

**Authors:** Ebtehal Alsolme, Saleh Alqahtani, Musa Fageeh, Duna Barakeh, Nitesh K. Sharma, Serghei Mangul, Heather A. Robinson, Amany Fathaddin, Charlotte A. E. Hauser, Malak Abedalthagafi

**Affiliations:** 1Genomic Research Department, King Fahad Medical City, Riyadh 12231, Saudi Arabia; ealsolme@kfmc.med.sa (E.A.); donabarakah@gmail.com (D.B.); 2Royal Clinic and Hepatology Department, King Faisal Specialist Hospital and Research Center, Riyadh 11564, Saudi Arabia; salalqahtani@kfshrc.edu.sa; 3Pathology Department, King Saud Medical City, Riyadh 12746, Saudi Arabia; m.fagih@ksmc.med.sa; 4The Titus Family Department of Clinical Pharmacy, School of Pharmacy, University of Southern California, Los Angeles, CA 90007, USA; prince26121991@gmail.com (N.K.S.); serghei.mangul@gmail.com (S.M.); 5Health eResearch, University of Manchester, Manchester M13 9PL, UK; robinsonheathera@googlemail.com; 6Department of Pathology, Collage of Medicine, King Saud University, Riyadh 11362, Saudi Arabia; afathaddin@ksu.edu.sa; 7Laboratory for Nanomedicine, Biological & Environmental Science & Engineering (BESE) Division, King Abdullah University of Science and Technology (KAUST), Thuwal 23955, Saudi Arabia; charlotte.hauser@kaust.edu.sa; 8Computational Bioscience Research Center (CBRC), King Abdullah University of Science and Technology (KAUST), Thuwal 23955, Saudi Arabia; 9Department of Pathology and Laboratory Medicine, Emory School of Medicine, Atlanta, GA 30307, USA

**Keywords:** BRCA2, PIK3CA, KRAS, colorectal cancer, somatic mutations, NGS, Saudi population, Saudi genome

## Abstract

Purpose: Next-generation sequencing (NGS) technology detects specific mutations that can provide treatment opportunities for colorectal cancer (CRC) patients. Patients and Methods: We analyzed the mutation frequencies of common actionable genes and their association with clinicopathological characteristics and oncologic outcomes using targeted NGS in 107 Saudi Arabian patients without a family history of CRC. Results: Approximately 98% of patients had genetic alterations. Frequent mutations were observed in *BRCA2* (79%), *CHEK1* (78%), *ATM* (76%), *PMS2* (76%), *ATR* (74%), and *MYCL* (73%). The *APC* gene was not included in the panel. Statistical analysis using the Cox proportional hazards model revealed an unusual positive association between poorly differentiated tumors and survival rates (*p* = 0.025). Although no significant univariate associations between specific mutations or overall mutation rate and overall survival were found, our preliminary analysis of the molecular markers for CRC in a predominantly Arab population can provide insights into the molecular pathways that play a significant role in the underlying disease progression. Conclusions: These results may help optimize personalized therapy when drugs specific to a patient’s mutation profile have already been developed.

## 1. Introduction

Colorectal cancer (CRC) is a malignant tumor of the large intestine (colon and rectum). In CRC, a gradual accumulation of genetic and epigenetic changes results in the transformation of normal colonic mucosa into invasive cancer [1]. It is the third most common cancer worldwide and the second highest cause of cancer-related deaths in most Western countries [2]. North America, northern and western Europe, and Australia have high incidence rates of CRC [3]. On a global scale, CRC accounts for 9.4% and 10.1% of all cancers in men and women, respectively. In the Gulf Cooperation Council (GCC) countries, CRC is the second most common cancer among both genders, with a reported 2.3-fold and 2.7-fold increase in recent years in newly diagnosed CRC cases among males and females, respectively [4]. In Saudi Arabia (SA), CRC cases were last reported at a frequency of 10.1% in men and 9.3% in women, which are close to global prevalence rates [5]. Currently, screening for CRC involves the detection of early-stage CRCs and pre-cancerous lesions in asymptomatic people, before it advances to later states and the patients are rendered ineligible for treatment [6]. In SA, despite its increasing incidence rates of CRC, there are no national screening policies for CRC [7,8,9,10,11]. Protocols that aid rapid detection and diagnosis of CRC in SA are therefore urgently required.

CRC presents as sporadic, inherited, or familial cancer. Sporadic CRC accounts for 70–75% of all diagnosed CRCs and is characterized by the absence of family history. Early-onset CRC cases in the Arab population are thought to be sporadic, and necessitate the evaluation of the primary molecular mechanisms and environmental factors. Inherited CRC accounts for 5–10% of all diagnosed CRCs [12]. Patients are often diagnosed with inherited syndromes such as familial adenomatous polyposis (FAP), *MUTYH*-associated polyposis (MAP), and hamartomatous polyposis syndromes, along with non-polyp-associated syndromes, such as Lynch syndrome (HNPCC), which increase the risk of development of CRC. Lastly, the least-understood pattern is known as “familial” CRC, accounting for ~25% of cases. In this category, CRC patients have a family history of the disease, but there is no clear pattern that is consistent with any of the known inherited syndromes [13].

The molecular assessment of malignant tumors is an important tool in the understanding of molecular pathways involved in the disease. With the advent of precision cancer therapy and personalized medicine [14], molecular profiling of tumors unravels information on patient diagnosis and prognosis that can be predictive of a successful therapeutic outcome [15,16,17]. Next-generation sequencing (NGS) has enabled the exploration of somatic-protein-altered mutations for many cancer types. Data regarding missense mutations within coding genes have been intensively accumulated [8,17,18]. NGS studies have led to the discovery of novel mutations, altered genes, and genomic rearrangements that have been used to evaluate CRC tumor response to standard therapy [14]. As an example, deregulation of the VEGF receptor and platelet-derived growth factor (PDGF) receptor is now known to be associated with CRC tumor progression and metastasis [19,20]. Accordingly, treatment with anti-VEGF drugs that inhibit angiogenesis, especially bevacizumab and ramucirumab, has improved therapeutic outcomes in metastatic CRC [2,21]. Anti-epidermal growth factor receptor (EGFR) agents, including cetuximab and panitumumab, in combination with chemotherapy, may also improve the survival of CRC patients with wild-type RAS tumors, but are ineffective in CRC tumors containing RAS mutations [15,17,22,23,24].

The Oncomine™ Comprehensive Assay v3 (OCAv3) covers 161 cancer-associated genes, allowing the detection of single nucleotide variants (SNV), multiple-nucleotide variants (MNV), and small insertions/deletions (indel). The OCAv3 has been routinely implemented in some clinical settings to assist oncologists’ decisions on therapeutic courses. The performance of OCAv3 has recently been used to focus treatment options for refractory metastatic colorectal cancer [25]. Developing a comprehensive, robust, accurate diagnostic tool for CRC requires an in-depth knowledge of population variants to distinguish disease-related mutations from rare variants without functional consequences. In the absence of such comparisons, NGS may give rise to false positive or negative results, resulting in incorrect decisions for clinical management and treatment regimens (false positives) or in undiagnosed conditions (false negatives) [21]. NGS approaches can prevent such errors by covering unexplored mutations in non-coding regions. Establishing a reliable NGS analytical system that considers variations specific to ethnic and population subgroups, where comparisons are drawn to healthy populations of the same ethnicity, is critical for accurate analysis [26,27,28,29].

Here, we conducted a cohort study involving sporadic CRC cases. We report somatic mutations in Saudi Arabian CRC cases, determined using targeted sequencing. We also describe the mutational profile of patients with CRCs using a targeted NGS approach and analyze their potential correlations with clinicopathological factors. In addition, we aimed to assess the biological and clinical significance of low variant allele frequency (VAF) for small variants and to compare them with those of The Cancer Genome Atlas (TCGA), a publicly available archive.

## 2. Materials and Methods

### Sample Selection

A total of 107 tissue samples were collected from King Fahad Medical City (KFMC) and King Abdullah University Hospital (KAUH). All clinical data were retrieved from electronic medical records. Essential demographic and disease-specific characteristics were extracted after the complete anonymization of data. Archived pathology specimens were reviewed by a board-certified pathologist (MF). CRC areas with a high tumor cell content (at least 70%) from unstained formalin-fixed paraffin-embedded (FFPE) tissue specimens were obtained for microdissection. The inclusion criteria were: (a) Saudi Arabian patients, (b) patients with no known genetic predisposition to CRC, (c) patients who had not received neoadjuvant chemotherapy, and (d) pathologically confirmed adenocarcinoma of the colon or rectum. DNA and RNA were extracted using RecoverAll™ Total Nucleic Acid Isolation Kit for FFPE, and the concentration was quantified using the Qubit™ ds High-Sensitive Assay kit on the Qubit fluorometer. All library preparation was performed manually according to manufacturer’s instructions. Multiplex PCR amplification was conducted using a DNA concentration of approximately 20 ng. For sequencing, prepared libraries were loaded according to manufacturer’s instructions and prepared using the Ion Chef™ System. Sequencing was performed using the Ion S5™ XL Sequencer. The data was mapped to human genome assembly 19, embedded as the standard reference genome in the Ion Reporter™ Software 5.18. Workflow Version: 4.2, which was used for initial automated analysis. Additionally, coverage analysis reports from the Ion Reporter™ Software providing measurements of mapped reads, mean depth, uniformity and alignment over a target region were used for quality assessment of the sequencing reactions.

## 3. Results

### 3.1. Cohort Demographics and Clinical Management

This study was reviewed and approved by the Institutional Review Board of KFMC. In total, 107 patients with CRC were included in this study. FFPE tissues for colorectal cancer (CRC) were collected from the Department of Pathology at KFMC, King Saud University (KSU), and King Saud Medical City (KSMC). The detailed clinicopathological characteristics of the study cohort are shown in Table 1. Therapy administered to patients constituted neoadjuvant chemoradiotherapy, and only one of the patients received regorafenib.

### 3.2. Mutational Profile Analysis

Of the patients studied, 87% (93/107) had somatic mutations, which were more frequent in the *BRCA2* (79%), *CHEK1* (78%), *ATM* (76%), *PMS2* (76%), *ATR* (74%), and *MYCL* (73%) genes. Genes with a mutation frequency > 1% are presented in Figure 1. Mismatch repair and *APC* screening were not performed in our NGS panel. Compared to the mutation frequencies reported in the TCGA CRC dataset, the mutation frequency in *TP53* (72%) was higher in our cohort, the frequencies of *PIK3CA* (25%) and *FBXW7* (21%) mutations were relatively equal, and the frequency of mutation in *KRAS* (37%) was relatively lower. 

### 3.3. BRCA2 Mutation

Approximately 80% (84/107) of the patients carried *BRCA2* mutations, showing 352 different variants. The missense mutation c.7397T>C (p.Val2466Ala), was the most common mutation and was observed in 78 patients (74.3%). Studies of breast and ovarian cancers have described this mutation as benign [30], and its clinical significance remains unknown. The second most common *BRCA2* mutation in this cohort was the *BRCA2* N372H non-conservative amino acid substitution polymorphism (asparagine to histidine substitution) as shown in Figure 2. This mutation has been associated with an elevated risk of overall cancer in predominantly Caucasian and African cohorts, with specific relationships already characterized in non-Hodgkin lymphoma and ovarian cancer [31]. This mutation was present in 52 patients (49.5%, Figure 2), and is the only recognized common non-synonymous polymorphism in the *BRCA2* gene [32].

In contrast, in the TCGA dataset, *BRCA2* was not among the top 50 most frequently mutated genes. Approximately 13% of the samples in the TCGA dataset had mutations in the *BRCA2* gene, accounting for 53 mutations. Interestingly, the mutations observed in our cohorts were not identified in the TCGA dataset. Studies have identified *BRCA1* and *BRCA2* variants among early-onset CRC (1.3%) [33], high-risk CRC (0.2%) [34], and unselected CRC patients (1.0%) [35] at a higher frequency. However, definitive proof of causality between CRC and its association with *BRCA1* and *BRCA2* pathogenic variants has not been established in the existing literature.

### 3.4. TP53 Mutations

A total of 96 different *TP53* variants were detected among 77 patients (73.3%), showing key differences from those observed in the TCGA dataset (55%). Among the most common polymorphisms were those affecting proline 72 (p.Pro72Arg *n* = 66; p.Pro72Cys *n* = 1). p.Pro72Arg has previously been studied, but it did not appear to predict the risk of colon cancer in the Iranian Azeri population [36], and p.Pro72Cys appears to be similarly benign. Variants were also seen in arginine residues 273 and 282 (p.Arg273His *n* = 4; p.Arg282Trp *n* = 4), which fall within a known hotspot in a *TP53* DNA binding domain. These mutations are associated with Li-Fraumeni syndrome [37]. We observed only a single case of p.Arg282Gln (Figure 3), which is pathogenic and has been observed at high frequency in other colorectal cancer cohorts [38].

In the TCGA data, the most common variants in *TP53* were p.Arg175His (*n* = 17) and p.Arg213Ter (*n* = 13), both associated with Li-Fraumeni syndrome [37], p.Arg248Trp (*n* = 13), which falls within the same hotspot as p.Arg273His and p.Arg282Trp, and p.Arg273His (*n* = 12), as in our own cohort (supplementary). p.Pro72Arg is one of the more than 200 single-nucleotide polymorphisms (SNPs) reported at the *TP53* locus, with studies reporting inconsistencies in the association between this SNP and increased risk of cancer. 

p.Arg175His, p.Arg248Trp, and p.Arg273His are three of the eight hotspot mutations (germline and somatic) reported in *TP53* that have been shown to have an increased likelihood of presentation with a classic Li–Fraumeni syndrome (LFS) phenotype, earlier age of first breast cancer onset, and shorter time to diagnosis of any cancer [39]. p.Arg175 and p.Arg273 have also been shown to play a critical role in submucosal invasion and metastasis of intestinal tumors through a gain-of-function mechanism [40].

### 3.5. KRAS Mutation

A total of 35 variants of *KRAS* were found in 40 (37.38%) patients, compared to 39.9% of patients in the TCGA dataset (Figure 4). The most common alterations were on the Gly 12 residue, which was substituted with Asp (*n* = 15), Ser (*n* = 5) Arg (*n* = 2), Val (*n* = 2), or Ala (*n* = 1). In recent studies of Saudi Arabian cohorts, 35–56% of patients carried *KRAS* mutations, placing our cohort towards the lower end of reported mutation frequencies for Arab cohorts [41]. Consistent with these studies, *KRAS*-G12D (glycine to aspartate) was the most common point mutation. Similarly, in the TCGA data, the Gly 12 residue was the most commonly altered residue, with variants observed in 102 patients, including substitutions with Asp (*n* = 48), Val (*n* = 33), Cys (*n* = 8), Ser (*n* = 7), Ala (*n* = 4), Arg (*n* = 1), and Phe (*n* = 1). The second most common altered residue was Gly 13, where eighteen and two cases showed substitutions with Asp and Cys, respectively, in the TCGA data, and three cases and one case showed substitutions with Asp and Ser, respectively, in our data.

*KRAS* codon 12 and 13 alterations are associated with colorectal liver metastasis [42], with an estimated 98% of *KRAS* mutations involving residues 12, 13, and 61 [43]. *KRAS* G12D mutations appear to lead to better overall survival (OS) rates than other *KRAS* mutations such as *KRAS-*G12C among CRC patients [44]. Our results did echo this correlation, although the difference between OS in patients with and without KRAS-G12D was not significant (*p* > 0.05).

### 3.6. PIK3CA Mutation

Mutations affecting the *PIK3CA* gene were found in 27 patients (25.7%), comprising 36 different variants (Figure 5). The most common mutation was in exon 9, including four cases of p.Glu545Lys. In the TCGA dataset, 29% of analyzed samples show simple somatic mutations where glutamic acid residue 545 was replaced by Lys, Ala, Gln, and Gly in 16, 4, 2, and 2 patients, respectively. Exon 9 (E545K) is the most common hotspot for *PIK3CA* mutation in CRC patients. In an Iranian CRC cohort, exon 9 mutations were associated with poorer survival, higher tumor stage, and greater tumor differentiation [45].

### 3.7. Survival Curves

Survival curves were plotted and stratified by categorical variables (AJCC stage, AJCC grade, presence or absence of pathogenic variant (PV) in each gene of interest (*BRCA2*, *TP53* or *KRAS*), presence of frequent specific *KRAS* mutations (*KRAS*-12D, *KRAS*-G12S, *KRAS*-G13D), tumor location, tumor grade, and TMB category) (Figure 6). A heat map of mutations was also constructed to visualize mutation frequency and show correlation in mutation frequencies between different genes (Figure 7). None of the univariate survival curves showed significant negative relationships between survival over time and categorical variables using log-rank tests (*p* > 0.05). However, the Cox proportional hazards model employed in our analysis revealed a rather unexpected and intriguing relationship between tumor grade and subsequent survival outcomes across time intervals. Specifically, the model indicated that poorly differentiated tumors were linked to a notably higher likelihood of survival over the defined time period (*p* = 0.025). 

This finding challenges conventional expectations, as one would typically assume that poorly differentiated tumors, indicative of a higher degree of malignancy, would correlate with a poorer prognosis and lower chances of survival. However, our results suggest a contrary trend, implying that certain factors associated with poorly differentiated tumors might actually be conferring a survival advantage over the specified time span.

This unexpected association requires further exploration and investigation into the underlying mechanisms at play. It underscores the complexity of interactions within the context of tumor biology and the potential influence of various factors on the survival trajectory of patients with poorly differentiated tumors. This discovery invites researchers and clinicians to delve deeper into the intricate interplay of molecular, genetic, and clinical variables that might contribute to this counterintuitive outcome, potentially offering novel insights into improving patient prognoses.

## 4. Discussion

This study describes the molecular basis of CRC in a cohort of Saudi Arabians using targeted NGS sequencing. The emergence of large, publicly available databases, such as the TCGA, with extensive genomic and epigenomic data provides a wealth of annotation resources for the comparison of population-based cohorts to identify genetic variants associated with CRC based on ethnicity. Although not included in our NGS panel, previous studies have revealed the role of *APC* in 96% of the cases evaluated in the Saudi Arabian population cohort, along with the *TP53* gene [46]. *KRAS* or *PIK3CA* mutations were significantly associated with poor survival in cases with wild-type *TP53* [47]. Other genes that have shown alterations in Saudi Arabian population cohorts are extensively reviewed by Younis et al. [41]. Deriving a list of common somatic mutations from our population cohort for which precision drugs have been developed will assist in the planning and development of treatment regimens for specific CRC patients (personalized therapy). For example, identifying patients with *KRAS* mutations could help to place patients on new precision drugs specifically developed for their genotypes, such as *KRAS*-G12C inhibitors AMG510 (sotorasib) [48] and MRTX849 (adagrasib) [42], and the EGFR inhibitor cetuximab, which shows a greater response rate in patients with the *KRAS-*G13D mutation [30]. Furthermore, microsatellite instability is considered an indication for immunotherapy and prescription of platinum drugs [49].

Mutations at residues 12 and 13 of *KRAS*, which are associated with CRC metastasis, occurred with significantly higher frequency in the present study (38% of cases) than in the TCGA dataset (13.11% and 7.52% respectively), but within the midrange of *KRAS* mutation frequencies observed in CRC cohorts worldwide [41]. AMG 510 is an inhibitory agent currently under development and may become available for patients with *KRAS-*G12C tumors. Preclinical studies have shown regression of *KRAS-G12C* tumors and improved antitumor efficacy of chemotherapy and targeted agents with AMG 510 [48]. The frequency of the *PIK3CA* E545K mutation was lower than that observed in the TCGA dataset (8.25% vs. 29.2%, respectively).

In our cohort, a majority of patients had either a high or very high tumor mutational burden (TMB), including 34 patients with high TMB (between 13 and 100 mu/Mb) and 55 patients with very high TMB (>100 mu/Mb). Exceptionally high TMB is indicative of the hereditary cancer syndrome associated with the inactivation of *MLH1, MSH2, MSH6, PMS2* or other genes. Although we excluded familial CRC cases in our cohort, patients with very high TMB results may require germline genetic testing [50]. The clinical utility of TMB remains a contentious issue, but it may be wise to look at cases with high TMB, as almost all colorectal tumors arising in patients with Lynch syndrome and sporadic CRCs have high TMB due to *MLH1* promoter hypermethylation [51,52].

Of specific note are mutations of the *BRCA2* gene, which was not in the top 50 mutated genes in the TCGA dataset but was positive in 80% (84/107) of the patients analyzed in this study and included 352 different variants. In the context of the TCGA dataset, mutations within *BRCA2* were detected in approximately 13% of the samples, resulting in a total of 53 distinct mutations. Intriguingly, the specific mutations we observed in our study’s cohorts were absent from the TCGA dataset. Existing studies have documented variations in *BRCA1* and *BRCA2* among early-onset CRC (1.3%), high-risk CRC (0.2%), and a general cohort of CRC patients (1.0%) at a relatively higher frequency. However, it is important to note that the current literature does not provide conclusive evidence establishing a direct causal relationship between CRC and the presence of pathogenic variants in *BRCA1* and *BRCA2.*

This is a surprising finding of our study, as the results of previous research on the association of BRCA mutations with the risk of CRC were negative or inconclusive. In a systematic review of 18 studies and a meta-analysis of 14 studies, the risk of colorectal cancer was shown to be moderately elevated in *BRCA1* (a 1.49-fold higher risk of CRC in *BRCA1* mutation carriers) but not in *BRCA2* mutation carriers [53]. However, in a recent meta-analysis of nine studies, no increase in colorectal cancer was found among patients carrying a BRCA mutation [54]. In addition, a comprehensive review of alterations in CRC in SA cohorts did not include the BRCA genes [41]. Reports link p.Val2466Ala, the most frequent mutation found in our cohort, with familial breast cancer in SA [55]. Based on this preliminary data, it is important to investigate the role of *BRCA2* in the Saudi Arabian population and specifically the CRC population, which might be the target of future studies.

The genes ataxia telangiectasia mutated (*ATM*) and ataxia telangiectasia and Rad3-related (*ATR*), both members of the PI3K family, are integral to the maintenance of chromosome integrity and genome stability. *ATR* identifies single-strand DNA breaks induced by UV radiation that proceed to phosphorylate *CHEK1* (Ser345), triggering cell cycle arrest and inhibition of DNA replication. The *ATM* gene, which produces a damage recognition protein, becomes activated in response to DNA damage caused by ionizing radiation or reactive oxygen. Once phosphorylated by ATM, the *CHEK2* gene orchestrates the activation of various proteins contributing to cell-cycle arrest, apoptosis, DNA repair, and centrosome duplication.

Existing research has pointed to the influence of SNPs within DNA repair genes, not only impacting an individual’s susceptibility to breast cancer [55,56], but also influencing lung cancer [57] and pancreatic cancer [58]. At the *ATM* locus, the presence of rs664677 and rs609429 in a homozygous state was linked to heightened breast cancer risk [59].

It is important to emphasize that despite the comprehensive knowledge in this domain, there is currently a gap in data exploring the potential relationship between genetic variants within the ATR-CHEK1 and ATM-CHEK2 pathways and their impact on susceptibility to colorectal cancer. In our study, we found elevated mutational frequencies of *CHEK1* (78%), *ATM* (76%), *ATR* (74%) among our cohort, which may further reveal the impact of these genes in CRC.

In the case of the *TP53* gene, although a higher percentage showed mutations in our cohort versus the TCGA cohort (73.3% vs. 55% respectively), the most common two SNPs were those affecting proline residue 72 (p.Pro72Arg *n* = 66; p.Pro72Cys *n* = 1), which are both considered benign [36],. Additionally, other SNPs in the DNA binding domain of *TP53* have been associated with Li-Fraumeni syndrome, and CRC patients with mutant p53 have been shown to have worse OS than those with WT p53 [60]. This difference in mutational frequency, as well as in the distribution in codons between our study and the TCGA data set, may be attributable to differences in the sample selection (exclusion of familial CRC cases) and ethnicity. Several cancer susceptibility genes have pleiotropic effects, increasing the risk of a spectrum of cancers to varying degrees [61]. *KRAS* mutations are known predictive markers of a negative response to EGFR inhibitors, such as cetuximab or panitumumab [62,63,64]. However, the prognostic role of *KRAS* mutations in disease-free survival and overall survival in CRC patients remains controversial.

In the analysis of survival data from our cohort, none of the survival curves showed significant differences based on one or more mutations in *BRCA2*, *TP53*, or *KRAS* and survival over time. This is probably because we did not differentiate patient variables, as described in Caucasian patient cohorts, from mutations that appear benign in these other contexts.

One of the key limitations of our study using the panels of mutations identified during the diagnosis and profiling of CRCs is the limited coverage of relevant genes. For example, the panel used here lacks analysis of the *APC*, *SMAD4*, and other genes that have been implicated in CRC and are frequently referred to as the driver genes [65]. In future studies, the choice of panel genes should have a strong basis, and already known polymorphisms in selected genes should be considered.

## 5. Conclusions

In conclusion, we describe the characterization of the molecular basis for CRC in SA using targeted NGS sequencing. We present a mutational landscape of actionable genes in CRC for the SA population in patients without a familial history of CRC. Additionally, we address the clinical relevance of low VAF variants. A comprehensive analysis of population-specific molecular markers for CRC can provide insights into the disease progression and pave the way for the drafting of recommendations and guidelines warranting the use of gene panels in routine diagnostic procedures.

## Figures and Tables

**Figure 1 diagnostics-13-02993-f001:**
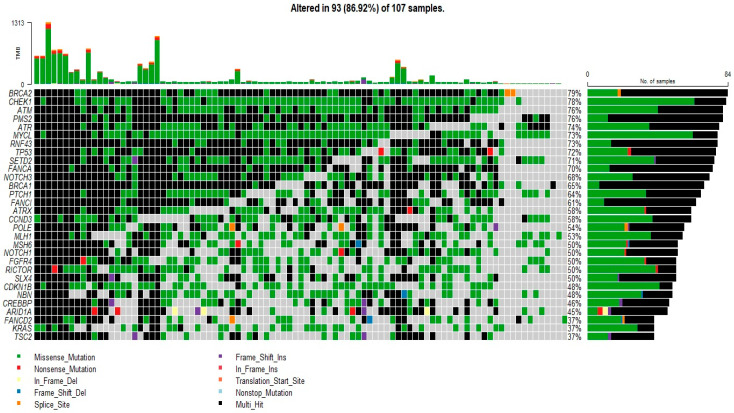
Mutation profile of patients with CRC. Oncoplot of the top 30 most frequently mutated genes in 107 patients. The figure lists genes with a mutation frequency >1%.

**Figure 2 diagnostics-13-02993-f002:**
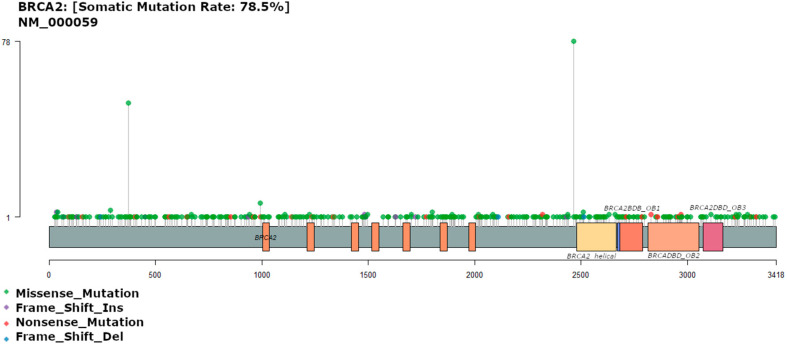
Gene map of *BRCA2* showing mutation rate hotspot loci. The gene map shows the mutation profile of 107 patients and the affected genes. The genomic profile was altered in 93 of the 107 patients analyzed. The figure lists genes with a mutation frequency of 1%.

**Figure 3 diagnostics-13-02993-f003:**
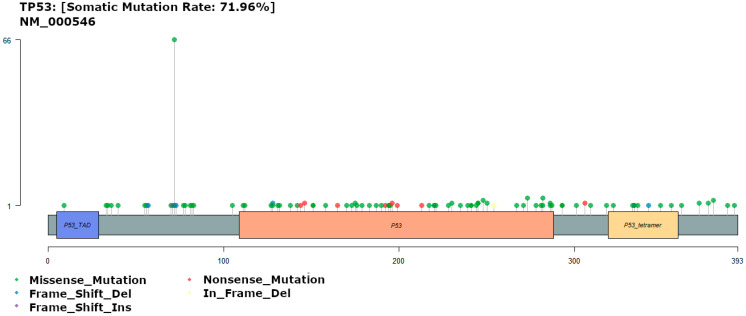
Gene map of *TP53* showing mutation rate hotspot loci. The gene map represents the mutation profile of 107 patients and the affected genes. In all, the genomic profile was altered in 71.9% of the patents. The figure lists genes with a mutation frequency >1%.

**Figure 4 diagnostics-13-02993-f004:**
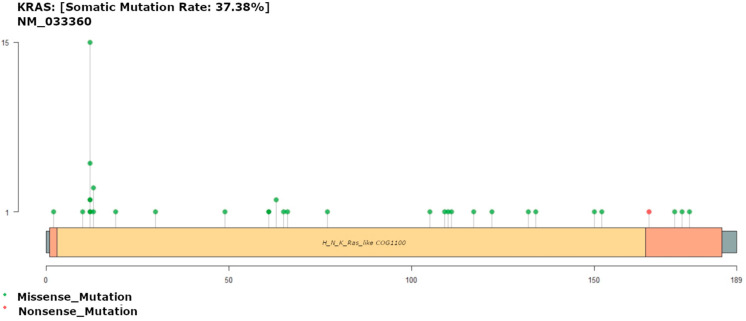
Gene map of *KRAS* showing mutation rate hotspot loci. The gene map represents the mutation profile of 107 patients and the affected genes. In all, the genomic profile was altered in 37.3% of the patients. The figure enlists genes with a mutation frequency of >1%.

**Figure 5 diagnostics-13-02993-f005:**
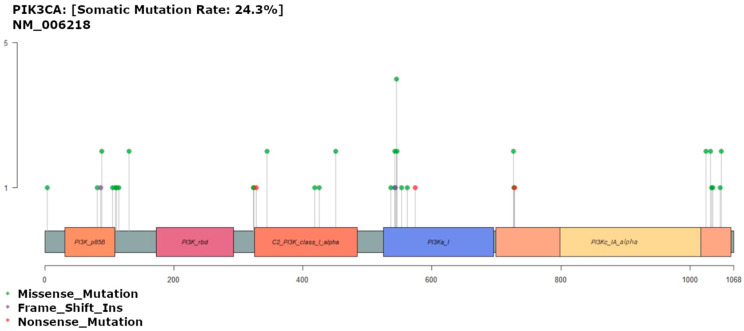
Gene map of PIK3CA showing mutation rate hotspot loci. The gene map shows the mutation profile of 107 patients and the affected genes. The genomic profile was altered in 24.3% of the patients. The figure lists genes with a mutation frequency >1%.

**Figure 6 diagnostics-13-02993-f006:**
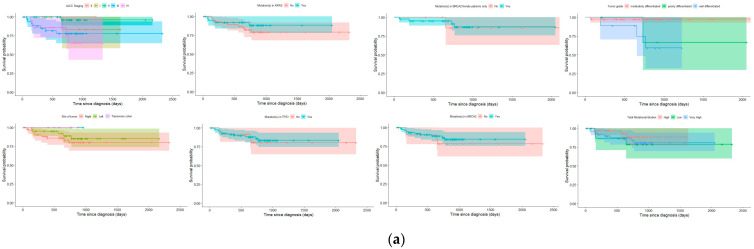
(**a**) Kaplan–Meier survival curves stratified by categorical variables (AJCC staging, AJCC grade, presence or absence of PV in each gene of interest (*BRCA2*, *TP53*, or *KRAS*). (**b**). Kaplan–Meier survival curves for specific *KRAS* mutations (*KRAS-*12D*, KRAS-*G12S, *KRAS-*G13D), tumor location, tumor grade, and TMB category.

**Figure 7 diagnostics-13-02993-f007:**
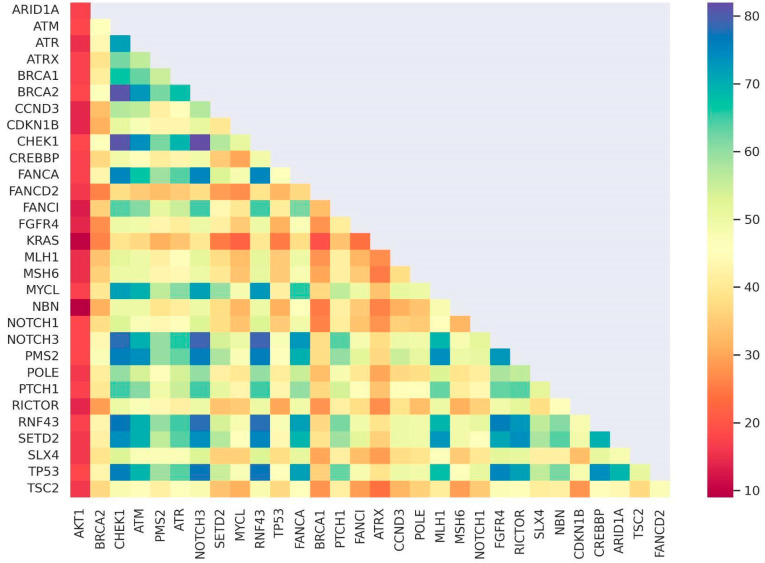
Heat map showing potentially correlated genes harboring mutations.

**Table 1 diagnostics-13-02993-t001:** Clinicopathological characteristics of the 107 CRC patients.

Gender	Female	F = 55
	Male	M = 52
Age (yrs)	Mean (sd)	58 (14.5)
	Range	95–20
Pathological Diagnosis	Adenocarcinoma	106
	Unknown CRC	1
Location	Right colon	56
	Left colon	45
	Transverse colon	6
Histological grade	Well-differentiated	16
	Moderately differentiated	84
	Poorly differentiated	6
	Unknown	1
Microvascular invasion	Present	30
	Absent	76
	Unknown	1
AJCC Stage	0	1
	1	10
	2	36
	3	36
	4	22
	Unknown	2

## Data Availability

The datasets presented in this article are not readily available because of the restriction in the Saudi Arabia law. Requests to access the datasets should be directed to the corresponding author and after getting approval from the NBEC https://ncbe.kacst.edu.sa/en/ (accessed on 20 July 2023).

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
