# Peer review of "The Genomic Landscape of Colorectal Cancer in the Saudi Arabian Population Using a Comprehensive Genomic Panel"

_diagnostics, 2023, doi:10.3390/diagnostics13182993_

Round 1

Reviewer 1 Report

The authors, analyzed mutation frequencies of common genes involved in colorectal cancer and their association with clinic-pathological characteristics and oncologic outcomes using NGS sequencing in 107 Saudi Arabian sporadic CRC patients.

They found most frequent mutations were BRCA2 (79%), CHEK1 (78%), ATM (76%), PMS2 (76%), ATR (74%), and MYCL (73%). However, no significant associations between specific mutations or overall mutation rate and overall survival were found.

Overall, this study did not provide any new finding related to molecular marker related colorectal carcinogenesis nor clinical characteristics. The common genes they found in this study are repeatedly reported in other countries in the world.

Author Response

Comments and Suggestions for Authors

The authors, analyzed mutation frequencies of common genes involved in colorectal cancer and their association with clinic-pathological characteristics and oncologic outcomes using NGS sequencing in 107 Saudi Arabian sporadic CRC patients.

They found most frequent mutations were BRCA2 (79%), CHEK1 (78%), ATM (76%), PMS2 (76%), ATR (74%), and MYCL (73%). However, no significant associations between specific mutations or overall mutation rate and overall survival were found.

Overall, this study did not provide any new finding related to molecular marker related colorectal carcinogenesis nor clinical characteristics. The common genes they found in this study are repeatedly reported in other countries in the world.

Reply to reviewer:

Thank you for taking the time to review our study. We sincerely appreciate your thoughtful assessment of our research on the mutation frequencies of common genes associated with colorectal cancer in Saudi Arabian sporadic CRC patients.

We have modified both result and discussion sections to highlight the findings our study provides:

Results

 The Cox proportional hazards model, employed in our analysis, revealed a rather unexpected and intriguing relationship between tumor grade and the subsequent survival outcomes across time intervals. Specifically, the model indicated that poorly differentiated tumors were linked to a notably higher likelihood of survival over the defined time period (p=0.025).

This finding challenges conventional expectations, as one would typically assume that poorly differentiated tumors, indicative of a higher degree of malignancy, would correlate with a poorer prognosis and lower chances of survival. However, our results suggest a contrary trend, implying that certain factors associated with poorly differentiated tumors might actually be conferring a survival advantage over the specified time span.

This unexpected association prompts further exploration and investigation into the underlying mechanisms at play. It underscores the complexity of interactions within the context of tumor biology and the potential influence of various factors on the survival trajectory of patients with poorly differentiated tumors. This discovery invites researchers and clinicians to delve deeper into the intricate interplay of molecular, genetic, and clinical variables that might contribute to this counterintuitive outcome, potentially offering novel insights into improving patient prognoses.

Discussion:
In the context of the TCGA dataset, mutations within the BRCA2 gene were detected in approximately 13% of the samples, resulting in a total of 53 distinct mutations. Intriguingly, the specific mutations we observed in our study's cohorts were absent from the TCGA dataset. Existing studies have documented variations in BRCA1 and BRCA2 among early-onset CRC (1.3%), high-risk CRC (0.2%), and a general cohort of CRC patients (1.0%) at a relatively higher frequency. However, it's important to note that the current body of literature does not provide conclusive evidence establishing a direct causal relationship between CRC and the presence of pathogenic variants in BRCA1 and BRCA2.

The genes Ataxia Telangiectasia Mutated (ATM) and Ataxia Telangiectasia and Rad3 Related (ATR), both members of the PI3Ks family, are integral to the maintenance of chromosome integrity and genome stability. ATR identifies DNA single-strand breaks induced by UV radiation, proceeds to phosphorylate CHEK1 (Ser345), triggering cell cycle arrest and inhibition of DNA replication. Whereas, the ATM gene, functioning as a damage recognition protein, becomes activated in response to DNA damage caused by ionizing radiation or reactive oxygen. Once phosphorylated by ATM, the CHEK2 gene orchestrates the activation of various proteins contributing to cell-cycle arrest, apoptosis, DNA repair, and centrosome duplication.

Existing research has pointed to the influence of single-nucleotide polymorphisms (SNPs) within DNA repair genes, impacting not only an individual's susceptibility to breast cancer (53,54) , but also influencing lung cancer (55) and pancreatic cancer (56). When it comes to ATM polymorphisms, the presence of rs664677 and rs609429 in a homozygous state was linked to heightened breast cancer risk [54].

Nevertheless, it's important to emphasize that despite the comprehensive knowledge in this domain, there is currently a gap in data exploring the potential relationship between genetic variants within the ATR-CHEK1 and ATM-CHEK2 pathways and their impact on susceptibility to colorectal cancer. In our study, we found this higher mutation among CHEK1 (78%), ATM (76%), ATR (74%).

Reviewer 2 Report

Summary: In the article, “The Genomic Landscape of Colorectal Cancer in the Saudi Arabian Population Using a Comprehensive Genomic Panel’, authors have use NGS to analyze the gene mutations associated with CRC in Arabian population.  The insights provided are significant and can be useful for recommendations and guidelines related to gene panels in diagnostics as concluded by the authors. The study design and result support the conclusions.

1.       The manuscript states “strong history of cancer….” in the inclusion criteria (b) in the Sample section of the materials and methods. The authors should consider explaining what they mean by strong history here.

2.       Also, for Figures 2-5 the authors should consider increasing the font size or some other modification to ensure the gene name and other details in the figure are easy to read. 

Author Response

Comments and Suggestions for Authors

Summary: In the article, “The Genomic Landscape of Colorectal Cancer in the Saudi Arabian Population Using a Comprehensive Genomic Panel’, authors have use NGS to analyze the gene mutations associated with CRC in Arabian population.  The insights provided are significant and can be useful for recommendations and guidelines related to gene panels in diagnostics as concluded by the authors. The study design and result support the conclusions.

Reply to reviewer

Thank you for providing positive feedback, We greatly appreciate your thoughtful consideration of our work.

Comment

  1. The manuscript states “strong history of cancer….” in the inclusion criteria (b) in the Sample section of the materials and methods. The authors should consider explaining what they mean by strong history here.

Reply to reviewer

We have removed this line from the text.

  1. Also, for Figures 2-5 the authors should consider increasing the font size or some other modification to ensure the gene name and other details in the figure are easy to read.

Reply to reviewer

Figure fonts have been modified, gene names and other details are enlarged now in the figures.
